# When Is the Right Moment to Pick Blueberries? Variation in Agronomic and Chemical Properties of Blueberry (*Vaccinium corymbosum*) Cultivars at Different Harvest Times

**DOI:** 10.3390/metabo12090798

**Published:** 2022-08-26

**Authors:** Miljan Cvetković, Milana Kočić, Dragana Dabić Zagorac, Ivanka Ćirić, Maja Natić, Đurađ Hajder, Aleksandar Životić, Milica Fotirić Akšić

**Affiliations:** 1Faculty of Agriculture, University of Banja Luka, 78000 Banja Luka, Bosnia and Herzegovina; 2Innovation Centre of Faculty of Chemistry Ltd., University of Belgrade, 11000 Belgrade, Serbia; 3Faculty of Chemistry, University of Belgrade, 11000 Belgrade, Serbia; 4Faculty of Agriculture, University in Bijeljina, 56000 Bijeljina, Bosnia and Herzegovina; 5Department of Fruit Science and Viticulture, Faculty of Agriculture, University of Belgrade, 11080 Belgrade, Serbia

**Keywords:** fruit weight, total phenolic content, radical scavenging activity, polyphenols, sugars

## Abstract

Blueberries, which are recognized by their colored fruits and exquisite flavor and taste, are a great source of bioactive substances with potential functional properties. For the purpose of this study, the blueberry cultivars ‘Duke’, ‘Chandler’ and ‘Bluecrop’ were picked at four different times. The aim of the study was to compare the cultivars and determine the best time for picking fruits for table consumption and to produce berries that can be used as functional foods with elevated levels of bioactive compounds. According to principal component analysis (PCA), the most influential traits for distinguishing different times of harvest in the ‘Duke’ cultivar were sorbitol, glucose, sucrose, and turanose; for the cultivar ‘Chandler’, they were caffeic acid, aesculetin, and quercetin; for the ‘Bluecrop’, they were fructose, maltose, radical scavenging activity, and quercetin. Blueberry fruits aimed for table consumption were those harvested in the first two pickings of the cultivar ‘Duke’, in the first and third of the ‘Bluecrop’, and in the third picking time of the cultivar ‘Chandler’, due to the highest fruit size and very high level of sugar (mostly glucose and fructose). ‘Duke’ berries from the second and third harvest (high level of total phenolic content, radical scavenging activity, total anthocyanins, aesculin, quercetin, and isorhamnetin), ‘Chandler’ from the first and third (the highest *p*-hydroxybenzoic acid, aesculetin, caffeic acid, phloridzin, kaempferol, kaempferol 3-O-glucoside, quercetin 3-O-rhamnoside, rutin, and quercetin) and ‘Bluecrop’ from the third harvest (highest level of total phenolics, radical scavenging activity, quercetin, rutin, quercetin 3-O-glucoside, kaempferol, quercetin 3-O-rhamnoside, kaempferol 3-O-glucoside, and isorhamnetin) had the highest levels of health-promoting compounds.

## 1. Introduction

Blueberry (*Vaccinium corymbosum* L.) is grown commercially in about 30 countries [1], which confirm its great popularity with producers and consumers. According to FAOStat [2], the leading countries are the USA, with 294k tones, followed by Peru (180k tones) and Canada (146k tones). Due to the high nutritional value and balanced taste, but also because of the high price of the fruit on the market, blueberry production and consumption is continuously rising [3,4]. Alongside the fact that blueberries are popular in all kinds of supermarkets and are used as table fruits, they can be processed into products such as juice, yogurt, jelly, jam, candy, ice cream, wine, and dried fruit [5] or as encapsulated extracts [6]. Their production also has ecological implications since it protects the forest floor from erosion and contributes to the formation of humus [7].

Numerous factors, such as the cultivar, ploidy (*2n*, *4n*, *6n*), climate, soil, water availability, cultural practices, and degree of maturity, affect the fruit quality of blueberries [8,9,10,11,12]. The fruit weight in blueberries is a cultivar-dependent trait, but growing conditions and cultural practices have a great influence on the size of the berries, which ranges from 0.9 to 4.0 g [13,14,15,16,17,18,19]. Undoubtedly, the genotype, the time of the harvest, and growing conditions have the greatest influence on the content of soluble solids, total acids, and vitamins in blueberry fruits [20,21,22,23]. Sun et al. [24] determined that the amount of vitamin C increased seven times in blueberry during maturation, while Mikulic-Petkovsek et al. [25] reported that organic acid contents were reduced by ripening. The genotype, the degree of fruit maturity, and fruit quality at the time of the harvest is an important factor that determines the quality of the fruit and both storage length and shelf life, so it is important to pick the fruit when its development and maturity is optimal for handling and consumption [26].

Blueberry fruits mostly consist of water (84%), carbohydrates (9.7%), and proteins (0.6%) [1]. They are also rich in many beneficial compounds, such as sugars, organic acids (citric, malic, quinic), vitamins (A, B-group, C), minerals (P, K, Mg, Ca, Na, Fe, Zn, Mn), carotenoids, and dietary fibers, particularly pectin. Their caloric value is low and they contain no fat [27]. The most abundant sugars are glucose and fructose, whose accumulation in berries is influenced by genetics and climatic conditions (particularly the light intensity).

Besides the extraordinary taste and aroma, blueberries have gained much attention due to their high level of health-promoting compounds based on considerable quantities of different phytochemicals. This premium-class foodstuff is rich in antioxidants, which help to eliminate free radicals, which play a role in chronic disease. In that way, phenolics from blueberry fruits have anti-inflammatory, antimicrobial, antihypertensive, anti-allergy, and antidiabetic properties [28,29]. In addition, the fruits protect against cancer, stroke, and urinary tract disease [30]. The quantity and profile of polyphenols depend upon the cultivar, climatic factors, production system, ripeness, and storage. According to Kalt et al. [31], climatic and geographical factors influence antioxidant value, although it is hard to differentiate between those and other biotic and abiotic factors. The most abundant bioactive compounds are flavonoids, phenolic acids, tannins, and anthocyanins [32]. Flavan-3-ols and flavonols (catechin, epicatechin, myricetin, and quercetin), hydroxycinnamic acids (including caffeic acids, ferulic acids and coumaric acids), hydroxybenzoic acids (including gallic acids and procatchuic acids), pterostilbene, and resveratrol are stored in higher quantities in the blueberry fruits [32,33]. Regarding anthocyanins, blueberries are distinguished by the presence of malvidin, delphinidin, petunidine, cyanidine, and peonidine, with sugar fragments of glucose, galactose, and arabinose. Ścibisz and Mitek [14] claimed that malvidin and delfinidine are the main components and amount to nearly 75% of all detected anthocyanins in blueberry fruits. Besides fruits, blueberry leaves, which can be used as food and as pharmaceutical ingredients, are rich in anthocyanins, 5-caffeoylquinic acid, several flavonols, glycosides, catechin, epicatechin, and chlorogenic acid, thus preventing many degenerative diseases [34,35].

Thus, the goal of the present study was to characterize the changes in fruit quality characteristics during the harvest season in blueberry fruits. Likewise, our goal was to compare ‘Duke’, ‘Bluecrop’, and ‘Chandler’ blueberry cultivars and determine the best date for harvesting for the various elements. In that regard, harvest time could be adjusted to the needs of a single cultivar to enhance their bioactive potential.

## 2. Materials and Methods

### 2.1. Plant Material

For the purpose of this study, cultivars ‘Duke’ (D), ‘Bluecrop’ (BC), and ‘Chandler’ (C) were used. Two-year-old nursery plants from all three blueberry cultivars (‘Duke’, ‘Bluecrop’, and ‘Chandler’), which are used for panting orchards, were purchased from the certified nursery “Vrtni centar Drijen” (Zagreb, Croatia) in 2013, who guaranteed that plant material was propagated from a known nucleus plant with inspected phytosanitary status, claiming that it was true to varietal type (accompanied with original tag). A plantation of 0.5 ha was established in 2013 with two-year-old nursery plants with well-developed root systems. Due to the easier control of weeds with a more favorable water–air regime, plants were planted on raised banks. The banks were 30 cm high and 80 cm wide. The banks were fenced with planks and covered with wood chips. The rows were set in a north–south direction. The planting distance between rows was 3.0 m, with 1.5 m between the plants in a row. The orchard was covered with an anti-hail net. All agro-technical measures were regularly carried out in the orchard with the aim of achieving a stable and high yield and high fruit quality. The fertilization was done with organic and mineral fertilizers, both fertigation and foliar. As a stimulant of vegetative growth, fertigation with water-soluble “Agro life” fertilizer (formulations of NPK 12:52:5 and 20:20:20) was performed. Phosphorus fertilizers were applied three or four times per year. Magnesium and calcium were applied by fertigation, while calcium was mainly added by foliar application. Land maintenance was done by maintaining grass in the inter-row spaces; mowing was done regularly. Intra-weed removal was done by manual weeding around the bushes. The irrigation system with two lateral pipes was installed in the orchard and was used both for irrigation and fertigation.

Pruning was done twice a year, the first time in a late winter or early spring and the second (green/summer) performed immediately after the harvest in the second half of July, with the aim of stimulating the growth of annual shoots and the formation of flower buds for the following year. Plant protection was carried out in the orchard and included all treatment against all economically important pathogens and pests.

The research included five bushes of each variety. Bushes were selected by random sampling from all parts of the plot. When choosing the bushes, it was considered that they represent an average sample in the plantation in terms of their development and reproductive potential.

Blueberries were harvested at full maturity. To ensure that all samples were equally ripe, fruits were harvested when fruit skin color was completely blue, with lightness (L) values between 31.18 and 35.40, chroma (C) values between 3.58 and 5.48, and hue angle (h°) values between 53.19 and 81.35 for the Duke variety; lightness (L) values between 30.08 and 37.86, chroma (C) values between 4.83 and 5.98, and hue angle (h°) values between 34.36 and 70.68 for the ‘Bluecrop’ cultivar; and lightness (L) values between 28.74 and 29.92, chroma (C) values between 2.38 and 4.93, and hue angle (h°) values between 27.63 and 48.14 for the ‘Chandler’ cultivar. The first harvest date was on 14 June for the ‘Duke’ cultivar. The ‘Bluecrop’ was picked on 21 June, and ‘Chandler’ on 5 July. Berries were harvested weekly until the end of the season, with the last harvest date on 26 July (Table 1). The percentage of mature fruits represents the amount of ripened berries per bush in the particular week of harvest. The total sum during the 4 harvest periods (weeks) represents 100% of the harvested fruits during the entire harvest.

After harvesting, the fruits were stored in a manual (mobile) refrigerator and transferred to the laboratory of the Faculty of Agriculture, University of Banja Luka, where pomological traits were determined. The rest of the samples were frozen and transported to the Faculty of Chemistry, University of Belgrade, for chemical profiling.

### 2.2. Pomological Traits

Fruit weight, height, and width were determined on a sample of 50 randomly picked fruits from five bushes per cultivar. Determination of fruit weight was performed on an analytical balance (Kern, EMB600-2, Merck KGaA, Darmstadt, Germany). The fruit shape index was calculated based on the ratio of fruit height and width. The percentage of soluble solids in blueberry juice (% Brix) and acidity (%) was determined using a rapid method with a digital refractometer (Atago Pal Bx/Acid 14).

Soluble solid/total acid ratio was determined as the ratio between average soluble solids and average total acids. Determination of the pH value of blueberry juice was performed using a digital pH meter (Atago Dph-2).

### 2.3. Reagents and Standards

Standards of phenolic compounds, as well as sorbitol, turanose, and Trolox were from Sigma-Aldrich (Steinheim, Germany). From Merck (Darmstadt, Germany) were purchased Folin-Ciocalteu reagent, sodium carbonate, sodium hydroxide, sodium acetate trihydrate, methanol (HPLC grade), acetonitrile (MS grade), and formic acid (MS grade). Trehalose, fructose, sucrose, maltose, glucose, and maltotriose were purchased from Tokyo Chemical Industry, TCI, (Zwijndrecht, Belgium), while gentiobiose and isomaltose were obtained from Tokyo Chemical Industry, TCI, (Tokyo, Japan). From Supelco (Bellefonte, PA, USA) were purchased PTFE syringe filters (13 mm, 0.45 µm). Standard solutions were prepared with ultrapure water (TKA Germany MicroPure water purification system, 0.055 µS/cm).

### 2.4. Sample Preparation

The procedure of extraction of polyphenols from blueberry samples was described by [36]. Blueberry fruits (approximately 10 g) were homogenized and mixed with 50 mL methanol containing 0.1% HCl and stirred for 1 h on a magnetic agitator at room temperature.

The extraction procedure was repeated twice for each sample, and all supernatants were combined and evaporated to dryness in a vacuum evaporator (IKA RV8, IKA^®^—Werke GmbH & Co. KG, Staufen, Germany). The residue after evaporation was dissolved in methanol/water (60/40, *v/v*) to 50 mL. The extracts prepared in this way were used to determine TPC, TAC, and RSA values and the content of individual polyphenols. All extracts were filtered through a PTFE filter (0.45 μm) before analysis.

### 2.5. Total Phenolic Content (TPC)

A partially modified Folin-Ciocalteu method [37] was used to determine the content of total polyphenols [38]. Briefly, 0.5 mL of extract was mixed with 0.5 mL of ultra-pure water, and 2.0 mL of diluted Folin-Ciocalteu (10%, *v/v*) reagent was added. After 5 min, 2.5 mL of 7.5% sodium carbonate was added. The mixture was allowed to stand for 2 h, and the absorbance was determined at 765 nm. The calibration curve was constructed using standard gallic acid solutions (20–100 mg/L). The obtained TPC values were expressed as g gallic acid equivalent (GAE) per kilogram of frozen sample and were presented as mean values of three measurements ± SD.

### 2.6. Radical Scavenging Activity (RSA)

Antioxidant capacity was determined using the DPPH method previously described in [36]. Blueberry extracts (0.1 mL) were mixed with 4 mL of methanol solution of DPPH (71 µM). After incubation for 1 h at room temperature, the absorbance of the solution was measured at 515 nm. Trolox (100–600 µmol L^−1^) was used as a standard, and the results were expressed as milimoles of Trolox equivalents (mmol TE) per kg of frozen weight. All results were presented as mean values of three measurements ± SD.

### 2.7. Total Anthocyanin Content (TAC)

Determination of TAC was performed according to the pH-differential method described in the literature [39]. There is a balance between two forms of anthocyanins, colored and colorless, depending on the pH of the solution. The content of total anthocyanins was determined by measuring the absorbance of the extract at pH = 1 (KCl/HCl, 0.025 mol L^−1^) and pH = 4.5 (NaCH_3_COOH/CH_3_COOH, 0.4 mol L^−1^). Measurements were performed at two wavelengths, 510 nm and 700 nm. Anthocyanin amounts were calculated and expressed as grams of cyanidin-3-glucoside (cy-3-glu) per kilogram of frozen blueberry sample.

### 2.8. UHPLC–DAD MS/MS Analysis of Polyphenolic Compounds

Phenolic compounds in blueberry extracts were determined using a Dionex Ultimate 3000 UHPLC system (Thermo Fisher Scientific, Bremen, Germany) coupled to a TSQ Quantum Access Max triple-quadrupole mass spectrometer (ThermoFisher Scientific, Bremen, Germany). Chromatographic separation conditions (column used, composition of the mobile phase, gradient elution conditions, flow rate, injection volume) were as previously described in [40]. The software used for controlling the instrument was Xcalibur v.2.2. Commercial standards were used for comparison and identification of polyphenols. Peak areas were integrated to estimate the amounts of phenolic compounds and finally expressed as mg/kg.

### 2.9. HPAEC/PAD Analysis of Sugars and Sugar Alcohols (Ion Chromatography with Amperometric Detection)

The system for ion chromatography, DIONEX ICS 3000 (Dionex, Sunnyvale, CA, USA), consisting of a quaternary pump (Dionex, Sunnyvale, CA, USA), was equipped with autosampler ICS AS-DV 50 (Dionex, Sunnyvale, CA, USA) and coupled to a pulsed amperometric detector with a gold-working electrode and Ag/AgCl reference electrode. Carbohydrates were separated on a Carbo Pac^®^PA100 high-performance anion-exchange column (4 × 250 mm) (Dionex, Sunnyvale, CA, USA) set to 30 °C. The mobile phase consisted of 600 mM sodium hydroxide, 500 mM sodium acetate, and ultrapure water, and the composition was changed during the time of analysis, as reported in detail in our previous work [41].

### 2.10. Statistical Analysis

Pomological data were analyzed in Microsoft Office Excel by applying two-factorial analysis of variance (ANOVA). All values in tables for parameters are shown as arithmetic mean with standard error ($\bar{X}$ ± S_e_)_._ Significant interaction effects were identified, presented graphically, and further analyzed.

Tukey’s test was used to evaluate the pomological and chemical data and to detect significant differences (*p* ≤ 0.05) between the mean values. Tukey’s test was performed using the NCSS statistical program (https://www.ncss.com/, accessed on 15 June 2022). PCA was carried out employing the PLS_Tool Box software package for MATLAB (Version 7.12.0), and all data were group-scaled prior to PCA [38].

## 3. Results

### 3.1. Climatic Conditions

The blueberry orchard where the study was conducted was in the municipality of Srbac, Bosnia and Herzegovina (44.99 °N; 17.45 °E). The research was carried out during 2017, which saw a small amount of precipitation during the summer months (Figure 1).

During the months of June and July, when the harvest took place, the total amount of precipitation was 22.2 mm and 18.4m, respectively. During the month of May, a very small amount of precipitation (21.4 mm) was also recorded, which could affect the growth of the fruit and thus the size.

High temperatures and an almost complete absence of precipitation were recorded throughout the harvest period (Figure 2). The last rainfall of 16.0 mm was recorded in the period of 12–14 June, when the harvest started. More serious amounts of precipitation of 16.2 mm were registered in the period of 17–18 July, which coincides with the period of the end of the harvest.

Climatic conditions recorded during 2017 confirm the fact that modern blueberry cultivation is impossible without irrigation. On the other hand, the absence of precipitation during the harvest period is positive because of the absence of conditions for the development of fruit diseases, which can affect the quality of production.

### 3.2. Yield and Pomological Traits

Cultivars ‘Duke’ and ‘Chandler’ had a significantly higher number of fruits per bush compared to the cultivar ‘Bluecrop’ (Table 2). The coefficient of variation (Cv) in all tested cultivars indicated the relative uniform number of fruits per bush.

Compared to the cultivar ‘Duke’ and ‘Chandler’, cultivar ‘Bluecrop’ had a significantly lower number of fruits per bush and per unit area (Table 3).

According to the analysis, the highest average fruit weight (Table 4) was achieved in ‘Bluecrop’ in the first harvest date (1.81 g), and the lowest in the cultivar ‘Chandler’ in the fourth harvest date (1.09 g). Cultivars ‘Duke’ and ‘Chandler’ had a similar trend of fruit weight throughout the harvest, where the highest values of fruit weight were in the third harvest date, followed with a significant decrease in the fourth term. In the cultivar ‘Bluecrop’, the first two pickings gave satisfactory fruit weight, but the third and fourth term showed a sharp decrease. This cultivar showed the largest variation for this trait, where the difference between the largest and the smallest berries was 0.69 g. On average, the cultivar ‘Duke’ had the highest fruit weight (1.59 g) compared to the other two cultivars and on the first harvest date (1.57 g) compared to other picking times.

The highest fruit shape index was obtained in the cultivar ‘Chandler’ in the fourth harvest date (0.82) and the lowest in the cultivar ‘Duke’ in the first harvest date (0.68). Cultivars ‘Chandler’ and ‘Bluecrop’ showed a similar tendency of the fruit shape index. ‘Duke’ achieved the highest value of the fruit shape index in the second harvest date, with the decrease of this value afterwards. The cultivar ‘Bluecrop’ showed an increasing trend in the fruit shape index, with the highest values in the fourth harvest date. The ‘Chandler’ cultivar stood out with a higher value of the fruit shape index compared to the other cultivars, with an increasing trend in the later harvest dates. The highest fruit shape index in all three cultivars was at the latest picking time (0.75) and the lowest in the first harvest (0.72). The highest average soluble solids were determined in the cultivar ‘Bluecrop’ from the third harvest date (15.04%) and the lowest in the cultivar ‘Chandler’ from the fourth harvest date (10.55%). When it comes to the harvest dates, the highest soluble solids were the highest in the third harvest dates, regardless of cultivar (13.52%), which was significantly higher in comparison with the fourth harvest date (12.28%) and in comparison with the second (11.97%) and the first harvest date (11.39%). The content of total acids was the highest in the cultivar ‘Chandler’ from the second harvest date (1.45%) and the lowest in the cultivar ‘Bluecrop’ from the second harvest date (0.44%). For total acids, there no harvest that had a particular cultivar with the highest/lowest value, but the highest variation between different harvest dates was in the cultivar ‘Bluecrop’. The cultivar ‘Duke’ showed the least variation when it came to this parameter, with a decrease in average total acids (%) in later harvest dates. On average for all picking times, the cultivar ‘Duke’ showed the lowest total acids (0.705%), while ‘Chandler’ had the highest (1.035%).

According Table 4, the highest average value of soluble solids/total acids ratio was achieved in the cultivar ‘Bluecrop’ in the second harvest date (38.63) and the lowest in the cultivar ‘Chandler’ in the second harvest date (8.43).

The ‘Duke’ cultivar had the least variation in soluble solid/total acid ratio values, with an average increase of this parameter in later harvest dates. By comparing cultivars and harvesting times, ‘Bluecrop’ and the second picking could be underlined as the ones with the highest soluble solid/total acid ratio (20.39% and 20.72%, respectively). This ratio was the lowest in the cultivar ‘Chandler’ and for the first picking of all cultivars (12.625% and 14.96%, respectively). When it came to fruit juice pH value, the lowest value was in the cultivar ‘Bluecrop’ in the fourth harvest date (2.87) and the highest in the cultivar ‘Bluecrop’ from the second harvest date (3.33). The lowest average pH value was obtained in the cultivar ‘Chandler’ (2.99), and the highest in the cultivar ‘Bluecrop’ (3.09), regardless of the harvest dates. The lowest average pH value was present in the fourth harvest date (2.98) and the highest in the second harvest date (3.09), regardless of the cultivar.

The conducted ANOVA showed that the significant difference between the cultivars and between harvest dates were determined for almost all studied traits. The interaction cultivar × harvest date (Figure 3) was also statistically highly significant for fruit weight (Figure 3A), fruit shape index (Figure 3B), total acids (Figure 3C), and soluble solid/total acid ratio (Figure 3D), but not for fruit juice pH value.

### 3.3. Determination of TPC, RSA, and TAC

Total phenolic contents of the blueberry cultivars were dependent on the time of harvest, as shown in Table 5. The obtained results for all tested cultivars indicated that the content of total polyphenols decreased during the harvest time. The highest contents of total polyphenols were determined for the cultivar ‘Duke’ in the first and second harvest time (2.52 g GAE/kg frozen weight (FW)) as well as in the variety ‘Bluecrop’ in the first harvest time (2.51 g GAE/kg FW). These TPC values were statistically significantly higher than all other obtained TPC values (Tukey’s test). Generally, the lowest total phenolic contents were measured for the ‘Chandler’ cultivar (1.61–1.78 g GAE/kg FW). The results of radical-scavenging activity determined in blueberry extracts showed mostly uniform RSA values (Table 5). The highest value of antioxidant capacity was found in the ‘Bluecrop’ cultivar in the third harvest time (29.98 mmol TE/kg), but this RSA value was not statistically and significantly higher when compared with the RSA values of the ‘Bluecrop’ cultivar obtained for the other harvest dates. In general, the lowest values of antioxidant capacity with statistically insignificant oscillations by harvest dates were registered in the ‘Chandler’ cultivar (24.71–26.16 mmol TE/kg).

The highest value of total anthocyanins (Table 5) was determined in the cultivar ‘Duke’ in the second harvest time (1.75 g cy-3-glu/kg). This amount was a statistically and significantly higher when compared to all other cultivars and harvest dates. In general, the contents of total anthocyanins were relatively high in the cultivar ‘Duke’ in all harvest dates except the last one. Notably lower values of total anthocyanins were found in the ‘Bluecrop’ and ‘Chandler’ cultivars, without certain trends that could be related to the harvest date.

### 3.4. Phenolic Composition of Blueberry Extracts

Phenolic contents in blueberries harvested at different harvesting times were different (Table 6). A total of 14 polyphenolic compounds, mainly phenolic acids and flavonoids, were quantified. Among the phenolic acids in all investigated samples, caffeic acid was quantified in the range of 0.283 to 0.709 mg/kg FW. Wang et al. [42] proved that no caffeic acid was quantified in the ‘Duke’ blueberry cultivar and 0.02 mg/kg in the cultivar ‘Bluecrop’, grown in Jilin province in China. Okan et al. [43] stated that caffeic acid ranged from 0.3 to 0.7 mg/kg, which is in accordance with our results. Content of 5-O-caffeoylquinic acid in all four harvest times of ‘Duke’ and ‘Bluecrop’ cultivars was in range from 2.191 to 8.737 and from 3.202 to 9.982 mg/kg of frozen weight, respectively. The same content (3.241–7.584 mg/kg frozen weight) was found in samples of ‘Chandler’ blueberries, with the exception of second harvest time, in which the content was several times higher (34.793 mg/kg FW). 5-O-Caffeoylquinic acid (also known as chlorogenic acid) has potential as antidiabetic and antiobesity and is thought to have a positive effect on cholesterol level and hypertension [44]. In all samples of the cultivar ‘Duke’, no *p*-hydroxybenzoic acid and quercetin-3-O-rhamnoside were detected.

The highest content of *p*-hydroxybenzoic acid was found in the first harvest time of the ‘Chandler’ cultivar and the third harvest of the ‘Bluecrop’ blueberries and amounted to 11.048 and 8.238 mg/kg FW, respectively. Ellagic acid was detected only in ‘Chandler’ samples from the first and third harvest times and amounted to 0.708 and 0.298 mg/kg FW, respectively.

Flavonoids quantified in blueberry samples were quercetin, kaempferol, isorhamnetin, and their glicosides, but the most represented was quercetin, with an average value of 87.222 mg/kg FW for all three cultivars and different ripening stages. This value is in accordance with our results published earlier [39] for organically produced ‘Duke’ blueberries, where the concentration of quercetin was 53.69 mg/kg. Wang et al. [42] and Okan et al. [43] confirmed no quercetin in investigated samples of ‘Bluecrop’ and ‘Chandler’ or much lower values (2.112 mg/kg) compared to our results. Quercetin concentrations for individual cultivars ‘Duke’, ‘Chandler’, and ‘Bluecrop’ ranged from 63.995 to 119.136 mg/kg FW, 47.120 to 130.781 mg/kg FW, and 72.343 to 136.391 mg/kg FW, respectively. The highest values were determined in the first harvest time of the cultivars ‘Chandler’ and ‘Bluecrop’ and in third harvest for the ‘Duke’ samples. The second-most represented flavonol was quercetin-3-O-glucoside, with concentration up to 24.354 mg/kg FW.

### 3.5. Sugar and Sugar Alcohol Profiles

Besides aroma, fruit color, and firmness, sugar content is one of the main traits that influence fruit taste and one of the main benchmarks used in the evaluation of the nutritive value and fruit quality of blueberries. In most cases, sugar level is altered by the genotype, cultivation techniques, and pre-harvest conditions [45].

The amounts of quantified sugars and sugar alcohols are summarized in Table 7. The cultivar ‘Bluecrop’ had the highest level of quantified sugars (65.13 g/kg FW), while the cultivar ‘Chandler’ had the lowest (56.72 g/kg FW), as presented in Table 7. The first harvest time gave fruits with the highest level of sugars (68.70 g/kg FW), while fruits from the third harvest had the lowest level of sugars (57.08 g/kg FW). Glucose and fructose were the most abundant sugars in all three studied blueberry cultivars and in all four picking times, accounting from 92% to 97% (average of 49.8% for glucose and 46.12% for fructose) of all soluble sugars detected. The third most prevailing sugar was either sucrose or sorbitol depending on the cultivar and picking time. Sucrose was, on average, 2.09% of total sugars quantified, while sorbitol was 1.70%. In the first two picking times of the cultivar ‘Duke’, in the third picking of the cultivar ‘Bluecrop’, and in the fourth harvest time of the cultivar ‘Chandler’, sorbitol was most abundant than sucrose.

The lowest level of glucose was in the third harvest of the ‘Chandler’ cultivar (22.424 g/kg FW), fructose in the third harvest of the ‘Duke’ cultivar (15.159 g/kg), and both sucrose and sorbitol in the first harvest of the ‘Bluecrop’ (0.682 g/kg FW and 0.412 g/kg FW, respectively). The highest levels of the majority of the main sugars were in the first picking time of the ‘Duke’ cultivar (glucose was 44.471 g/kg FW, sucrose 2.267 g/kg FW, and sorbitol 3.781 g/kg FW). The exception was fructose, whose highest level was s in fruits of the cultivar ‘Bluecrop’ picked in the third harvest time (37.82 g/kg).

Trehalose, isomaltose, turanose, gentiobiose, maltose, and maltotriose were also detected in all cultivars and all harvest times but in small amounts. The only exception was gentiobiose, which was detected in the first three harvests of the ‘Duke’ cultivars and the fourth harvest of the ‘Chandler’ cultivar.

### 3.6. Principal Component Analysis

A principal component analysis (PCA) was used to distinguish differences among blueberry samples according to their chemical compositions. Quantified polyphenols, sugars, and sugar alcohols, together with TPC, RSA, and TAC values were used as variables (Appendix A). The initial matrix of 12 (the number of blueberry samples) × 27 (quantified polyphenols, quantified carbohydrates, TPC, RSA, and TAC) was processed using the covariance matrix with autoscaling. The cumulative variation of the dataset explained by the first three components was 69.7%. The first three principal components accounted for 39.1, 19.5, and 10.8% of total variability. Although there is no clear differentiation among blueberry samples, the PCA score plots for the first three principal components presented in Figure 4A1,B1,C1 show some trends and groupings. The PC1/PC2 score plot showed separation of ‘Duke’ in the first two harvest dates (D1 and D2) along the PC2 axis (Figure 4A1). Notably higher contents of several quantified sugars and sugar alcohols (sorbitol, glucose, sucrose, and turanose) are the most important factors responsible for the separation of samples D1 and D2 from the other blueberries (Figure 4A2). The ‘Chandler’ sample in the first ripening stage was separated from the all other samples according to higher contents of caffeic acid (7), aesculetin (8), and quercetin (15) (Figure 4A2). The ‘Duke’ sample (D3) was distinguished along the PC3 axis (Figure 4B1) by the highest amount of aesculin (Figure 4B2). The ‘Duke’ sample from the first harvest date (D1) was characterized by the highest content of glucose and was separated from the other blueberry samples along the PC2 axis (Figure 4C1). The highest content of glucose (20) on the PC2/PC3 score plot showed separation of the ‘Duke’ sample from the first harvest date (D1)

## 4. Discussion

High yields are the basis of profitable blueberry production. The yield per bush of the ‘Duke’ variety of 7.52 kg can be considered extremely high, bearing in mind the statements of Milivojević et al. [46], who determined the average yields of this cultivar as ranging from 3.5 to 7.4 kg/bush in a ten-year study. The yield depends to a significant extent on the size of the berries, so their abundance does not necessarily guarantee a high yield. Ehlenfeldt and Martin [17] stated that the small size of the berries is a consequence of a poor pollination, which leads to a decrease in yield. The same authors proved that small berries are formed due to an excessive number of shoots per plant and number of plants in the field. The method of cultivation can also have an influence on the yield. Milivojevic et al. [46] determined that the yield per bush of the cultivar ‘Duke’, which was grown under a hail net, was in the range of 5.2–6.0 kg/bush, while the plants that were grown in the open field gave a yield in the range of 4.1–4.8 kg/bush. The yield of the cultivar ‘Bluecrop’ in a slightly denser planting system was 3.3 kg/m^2^ for plants aged 3–5 years, as determined by Heiberg et al. [47]. In the later period of cultivation, the yield was significantly higher, which is the main reason why the authors recommend raising plantations with better-quality and better-developed nursery plants. 

The size of the fruit is primarily a genetically based characteristic, which depends to a significant extent on the conditions and method of cultivation. The highest average fruit weight of the ‘Bluecrop’ cultivar was achieved in the first harvest time (1.81 g), which confirms the statements of Castrejon et al. [15] and Zorenc et al. [4], who examined blueberry cultivars at five and four stages of maturation and ripening, respectively, and found that this cultivar has large berries and is highly favored by producers (easier manual harvesting) as well as consumers (attractive fruit size). A significantly lower berry weight was recorded in the cultivar ‘Bluecrop’ in the last two picking times. The decrease in the berry weight during successive picking can be related to the high temperatures and increased solar radiation at the end of the harvest, which affects the water status of plants and decreases the intensity of the photosynthesis process [48]. According to Sousa et al. [13] the decrease in fruit weight during the ripening period of the cultivar ‘Bluecrop’ was around 35.50%. The ‘Duke’ cultivar had a high and uniform fruit weight during the first three harvest times, while in the fourth term there was a slight decrease, as previously proved by Milošević and Milošević [49] and Milivojevic et al. [29]. The cultivar ‘Duke’ can form fruits weighing 2.29 g [50], which underlines the statement about the importance of the blueberry cultivation system. The ‘Chandler’ cultivar had smaller fruits, although Balkhoven et al. [16] stated that this cultivar can reach a weight of 2.5 g in Dutch conditions. In this study, blueberry fruits were slightly flattened (shape index 0.68–0.82). The shape of the fruit can be greatly influenced by the presence of the anti-hail net in the plantation [51].

The content of soluble solids recorded in this research ranged from 10.55% (for the ‘Chandler’ cultivar in the fourth harvest term) up to 15.04% (for the ‘Bluecrop’ cultivar in the third harvest term), which agrees with the authors who studied blueberries in Serbia [11], Romania [23], and Poland [14]. The cultivar has the greatest influence on the content of soluble solids in the fruits [19] in addition to the moment of ripening [14]. The variation of the content of soluble solids in fruits at different moments of harvest can be explained by the fact that the accumulation of soluble solids in berries is more intense if the weather is warmer and without precipitation immediately before and during their ripening [20,21].

The content of organic matter in berries, including the content of soluble solids, is also influenced by the age of the bush, the physiological and health status of the plant, the degree of ripeness of the berries, their position in the bush, and measures of plant care [52]. Iglesias et al. [22] reported that the soluble solid content decreased when black hail netting was used in the production. Bolling et al. [53] stated that the decrease in organic acid content from the first to the last harvest date affects the increase of soluble solid content. The content of total acids greatly affects the sweetness index and thus the taste of the fruit. Zorenc et al. [4] declared that the content of total acids in the ‘Bluecrop’ cultivar largely depends on the location where it is grown.

Previously, Mikulic-Petkovsek et al. [54] and Vance [19] proved that citric acid is the most abundant in blueberry fruits, followed by quinic acids. Together, they make up approximately 97% of the total organic acids in the blueberry. Malic acid is represented by less than 3% of total organic acids. Changes in the content of total sugars and organic acids during the blueberry harvest period in our research were consistent with the results of fruit ripening status reported by other researchers [55,56]. During this research, the recorded pH value range from 2.87 to 3.33 was in agreement with the statements of several authors [4,13,14,57,58].

Blueberry fruit contains notable amounts of anthocyanins, polyphenols, proanthocyanidins, and flavonoids, which are antioxidants of great importance for human health [42]. Numerous factors, such as ecological conditions (climatic and soil characteristics), cultivation technology, genotypic specificities, and maturity stage, can influence the total phenolic content in the high-bush blueberry [14]. The investigation of Castrejon et al. [15] showed that variations in the total phenol content in blueberries occur during the ripening process, and the TPC values are higher in the initial stages when compared to later stages of ripening.

However, the content of total phenols in our study was more influenced by variety than by the time of harvest. This is in accordance with the research of Ścibisz and Mitek [14], who showed that the cultivar has a great influence on the content of total polyphenols. The TPC values obtained in our study (1.45–2.52 g GAE/kg FW) agree with the results from our previous study (2.27–6.26 g GAE/kg FW) [45] and the literature data (2.08–4.96 g GAE/kg FW) [14]. As for RSA values, our findings (24.71–29.98 mmol TE/kg) are in line with the investigation of Ścibisz and Mitek [14] (23.1–40.4 mmol TE/kg). By analyzing numerous fruit species, Garcia-Alonso et al. [59] and Gündoğdu et al. [60] found that the total antioxidant capacity of blueberry is ranked third after persimmon and blackberry and followed by arbutus fruits. The total anthocyanin content accumulates in large quantities during the fruit-ripening phase [61]. Zorenc et al. [4] recorded similar values of total anthocyanins in the ‘Duke’ variety when compared to our findings. On the other hand, Kim et al. [62] noted significantly higher values of total anthocyanins in blueberries grown in Korea.

Regarding polyphenol profile and production of the functional food, the situation was complicated because phenolic compounds varied with maturation. Fruits of the cultivar ‘Duke’ from the second and third harvest had high levels of TPC, RSA, TAC, aesculin, quercetin, and isorhamnetin. In the ‘Chandler’ cultivar, fruits from the first picking time had high levels of *p*-hydroxybenzoic acid, caffeic acid, aesculetin, and quercetin and from the third period of harvest the highest level of quercetin 3-O-glucoside, kaempferol 3-O-glucoside, phloridzin, and kaempferol. In the ‘Bluecrop’ cultivar, berries from the third harvest period had high quantities of TPC, RSA, quercetin, rutin, quercetin 3-O-glucoside, kaempferol, quercetin 3-O-rhamnoside, kaempferol 3-O-glucoside, and isorhamnetin A similar situation with changing quantities of certain polyphenolic compounds during ripening was noticed in *Arbutus* fruits [63] and *Rosa* species [64]. According to Pedisic et al. [65] in cherry fruit and Liu et al. [66] in peach fruit, phenolic compounds decrease with ripening, while Usenik et al. [67] stated that phenolic compounds increase in plum fruits while ripening. In red currants, the most studied phenolic acids were higher in the green stage, while flavonoids were higher as ripening proceeded. Mikulic-Petkovsek et al. [25] reported that the flavanol value increased during ripening, while the flavonol value decreased in red currant.

Sugars play an important role in fruit flavor and thus consumer’s acceptance. In addition, sugars are energy sources in plants, important for fertilization, senescence, responses to all kinds of biotic and abiotic stresses, and synthesis of amino acids, polyphenols, and pigments [43]. The main sugars found in the studied blueberry cultivars were fructose and glucose, which is consistent with the findings of Ayaz et al. [68], Šne et al. [69], and Zhang et al. [5]. In our study, the glucose content (49.8%) was higher than the fructose content (46.12%) but both were higher than reported percentages in the studies of Šne et al. [69] and Zhang et al. [5].

In this study, glucose levels ranged from 22.424 to 44.471 g/kg FW and fructose levels ranged from 15.159 to 37.82 g/kg FW, which is lower than the blueberry cultivars grown in China and reported by Zhang et al. [5]. Since fructose is the sweetest sugar, the cultivar ‘Bluecrop’ could be recommended because its fruits on average had the highest levels of this pentose. The average sucrose concentration in all cultivars studied and in all harvest times was ~2%, which is probably due to the strong invertase activity during the ripening time [70].

Regardless of the harvest time, the ‘Duke’ cultivar had the highest level of glucose (34.20 g/kg FW) but lowest level of fructose (23.78 g/kg FW). ‘Bluecrop’ had fruits with balanced levels of glucose and fructose (32.13 g/kg FW and 31.04 g/kg FW, respectively). Nindo et al. [71] and Ayaz et al. [68] revealed that the amounts of fructose and glucose are mostly close to each other in blueberry fruits. This was also proved by Forney et al. [72], who claimed that equity between glucose and fructose was noticed even in white ‘turning blue’ fruits. The highest level of sorbitol and sucrose (1.78 g/kg FW and 1.53 g/kg FW, respectively) was seen in the fruits of the cultivar ‘Duke’. Disregarding cultivars, the fruits from the first harvest time had the highest level of glucose, fructose, sorbitol, and sucrose (33.32 g/kg, 31.91 g/kg FW, 1.64 g/kg, and 1.60 g/kg, respectively), while the lowest glucose (28.47 g/kg) was in the third harvest time and fructose (25.36 g/kg) in the fourth.

## 5. Conclusions

The results of this experiment confirmed the extraordinary quality of the blueberries grown in Bosnia and Herzegovina, which are reservoirs of sugars and compounds with high antioxidant properties. The optimum harvest time for table consumption was in the first two picking times for the cultivar ‘Duke’, first and third for the cultivar ‘Bluecrop’, and the third picking time for the cultivar ‘Chandler’ due to the highest fruit size, the very high level of sugar, and enough acids to ensure the fresh taste of the berries. Regarding the polyphenol profile, fruits of the cultivar ‘Duke’ from the second and third harvest time could be used for making functional food. In the ‘Chandler’ cultivar, the fruits that had the highest level of health-promoting compounds were those from the first and third period of harvest. In the ‘Bluecrop’ cultivar, berries from the third harvest period gave fruits with the highest bioactive compounds. Since harvest of the *Vaccinium corymbosum* fruits can last several weeks, this study helped us to improve our knowledge about chemical changes in blueberry fruits during ripening, which may help us to improve fruit quality and healthful properties. Overall, by designing the optimal harvest time of the studied blueberry cultivars, producers can offer and consumers can consume narrowly specialized *Vaccinium corymbosum* fruits. In that way, in the future, other fruit species with prolonged ripening and harvest times can be analyzed in order to optimize fruit quality and specify potential applications.

## Figures and Tables

**Figure 1 metabolites-12-00798-f001:**
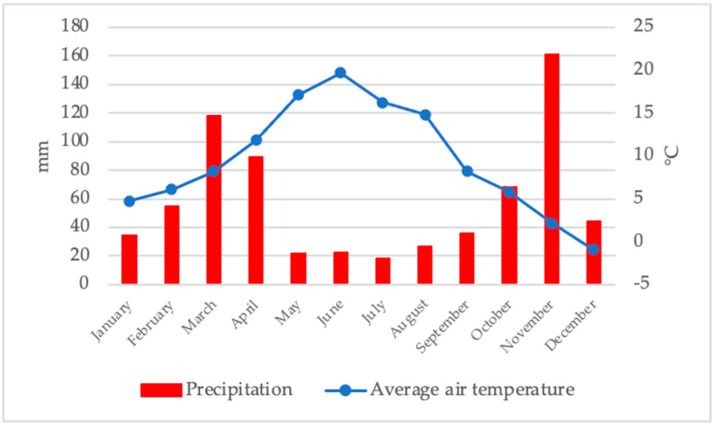
Climatic conditions (total rainfall and average air temperature) during 2017.

**Figure 2 metabolites-12-00798-f002:**
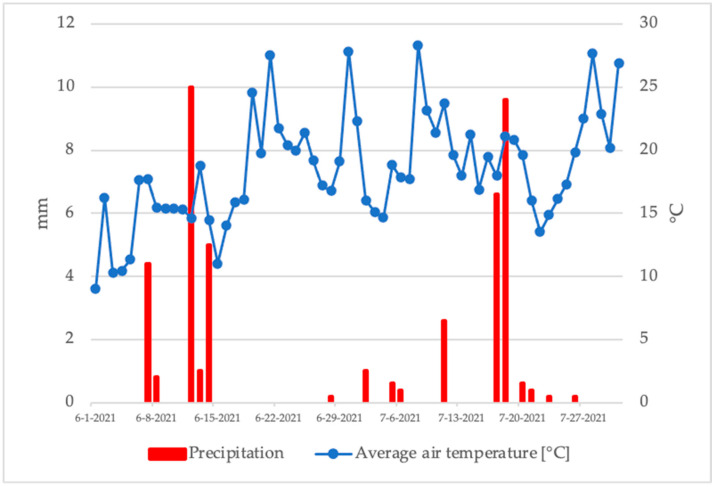
Daily precipitation and average air temperature during the harvesting period.

**Figure 3 metabolites-12-00798-f003:**
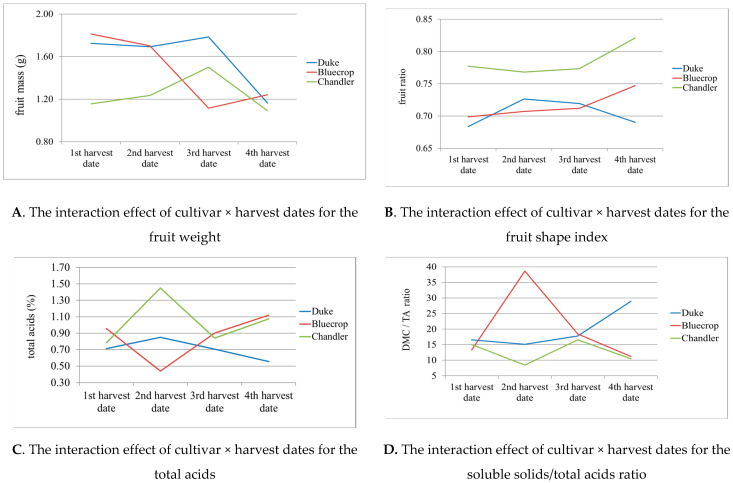
Significant interaction effect of cultivar × harvest date for fruit weight (**A**); fruit shape index (**B**); the content of total acids (**C**); and soluble solid/total acid ratio (**D**).

**Figure 4 metabolites-12-00798-f004:**
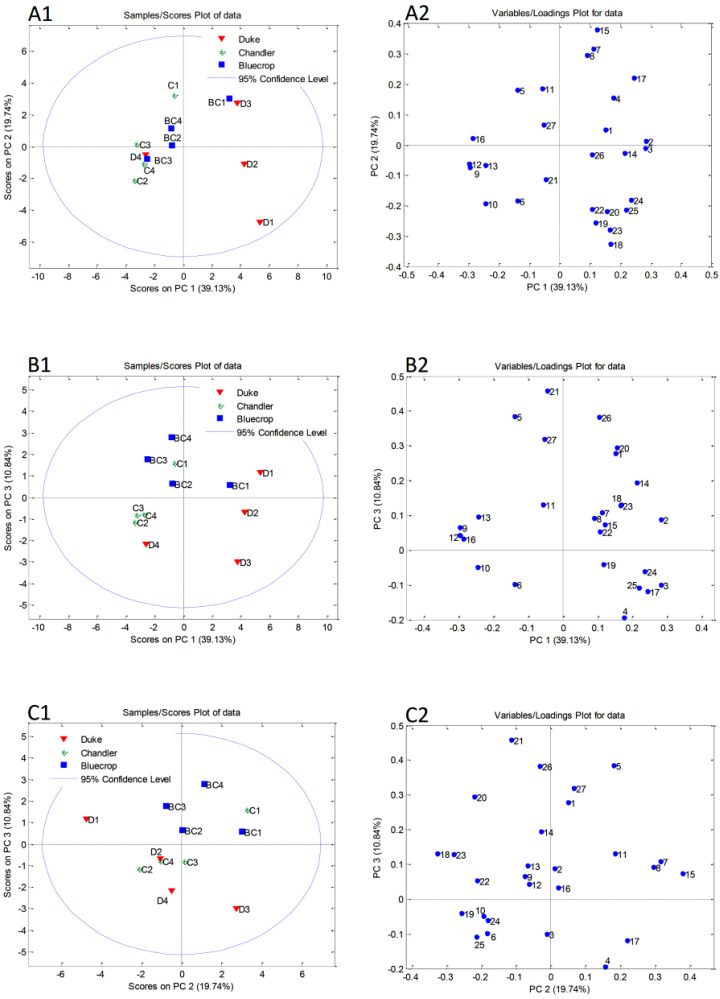
Principal component analysis (PCA) performed on chemical parameters obtained for blueberry samples: Score plots of the first three principal components (**A1**–**C1**) and loading plots (**A2**–**C2**). The number of the loading plots corresponds to the chemical properties given in Appendix A.

**Table 1 metabolites-12-00798-t001:** The harvest dates and percent of mature fruits (fruits ripe for harvest) during each harvest.

Cultivar × Harvest Date	Abbrevation	Harvest Dates	% of Mature Fruits
‘Duke’–1st harvest	D1	14 June 2021	10
‘Duke’–2nd harvest	D2	21 June 2021	20
‘Duke’–3rd harvest	D3	28 June 2021	50
‘Duke’–4th harvest	D4	05 July 2021	20
‘Bluecrop’–1st harvest	BC1	21 June 2021	10
‘Bluecrop’–2nd harvest	BC2	28 June 2021	15
‘Bluecrop’–3rd harvest	BC3	05 July 2021	35
‘Bluecrop’–4th harvest	BC4	12 July 2021	40
‘Chandler’–1st harvest	C1	05 July 2021	10
‘Chandler’–2nd harvest	C2	12 July 2021	20
‘Chandler’–3rd harvest	C3	19 July 2021	40
‘Chandler’–4th harvest	C4	26 July 2021	30

**Table 2 metabolites-12-00798-t002:** Average number of fruits per bush.

Cultivar	Bluecrop	Duke	Chandler
$\bar{X}$	2301.80	5331.40	6085.20
S$\bar{x}$	109.28	232.11	191.37
Cv	10.61	9.73	7.03

**Table 3 metabolites-12-00798-t003:** Average yield per bush (kg) and unit area (t/ha).

Variety	Yield (kg/bush)	Yield (t/ha)
Bluecrop	3.66	8.13
Duke	7.52	17.39
Chandler	7.59	16.87

**Table 4 metabolites-12-00798-t004:** Pomological traits in three different blueberry cultivars at four harvest dates.

Cultivar × Harvest Date	Fruit Weight (g)	Fruit Shape Index	Total Soluble Solids (%)	Total Acids (%)	Total Soluble Solids/Total Acids Ratio	Fruit Juice pH Value
D1	1.73 ± 0.05 de*	0.68 ± 0.0060 a	11.03 ± 0.22 ab	0.71 ± 0.13 ab	16.57 ± 3.13 ab	3.17 ± 0.07
D2	1.69 ± 0.05 d	0.73 ± 0.0035 bc	11.74 ± 0.87 bc	0.85 ± 0.17 ab	15.10 ± 3.31 ab	3.03 ± 0.07
D3	1.78 ± 0.01 de	0.72 ± 0.0052 b	12.56 ± 1.78 cd	0.71 ± 0.08 ab	17.83 ± 2.19 ab	3.03 ± 0.07
D4	1.16 ± 0.03 ab	0.69 ± 0.0054 a	13.82 ± 0.33 d	0.55 ± 0.13 a	28.93 ± 8.24 bc	3.07 ± 0.09
C1	1.16 ± 0.005 ab	0.78 ± 0.0060 d	11.62 ± 0.37 ab	0.78 ± 0.07 ab	15.03 ± 1.13 ab	3.03 ± 0.07
C2	1.24 ± 0.04 b	0.77 ± 0.0103 cd	12.20 ± 0.18 cd	1.45 ± 0.05 c	8.43 ± 0.31 a	2.90 ± 0.06
C3	1.50 ± 0.01 c	0.77 ± 0.0082 cd	12.98 ± 0.36 cd	0.84 ± 0.16 ab	16.57 ± 3.09 ab	3.03 ± 0.03
C4	1.09 ± 0.03 a	0.82 ± 0.0161 e	10.55 ± 0.41 a	1.07 ± 0.16 b	10.47 ± 2.17 a	3.00 ± 0.06
BC1	1.81 ± 0.04 e	0.70 ± 0.0180 ab	11.54 ± 1.05 ab	0.96 ± 0.18 b	13.28 ± 3.18 ab	3.00 ± 0.06
BC2	1.70 ± 0.05 d	0.71 ± 0.0033 ab	11.98 ± 0.71 bc	0.44 ± 0.18 a	38.63 ± 15.28 c	3.33 ± 0.26
BC3	1.12 ± 0.03 a	0.71 ± 0.0037 ab	15.04 ± 0.92 e	0.90 ± 0.21 b	18.47 ± 4.28 ab	3.17 ± 0.03
BC4	1.24 ± 0.02 b	0.75 ± 0.0108 c	12.46 ± 0.28 cd	1.12 ± 0.06 bc	11.18 ± 0.60 a	2.87 ± 0.03

* Different letters within the same column indicate statistically significant difference at *p* < 0.05 by Tukey’s test.

**Table 5 metabolites-12-00798-t005:** Total phenolic content, radical-scavenging activity, and total anthocyanin content of blueberry extracts.

Samples	TPC (g GAE/kg)	RSA (mmol TE/kg)	TAC (g cy-3-glu/kg)
D1	2.52 ± 0.02 a*	27.64 ± 1.46 bcde	1.65 ± 0.01 b
D2	2.52 ± 0.02 a	29.66 ± 0.00 a	1.75 ± 0.01 a
D3	2.35 ± 0.01 b	27.78 ± 0.51 cd	1.61 ± 0.02 c
D4	1.45 ± 0.02 i	24.83 ± 0.06 f	0.49 ± 0.00 k
C1	1.78 ± 0.01 f	25.91 ± 1.11 ef	0.97 ± 0.01 e
C2	1.64 ± 0.02 h	24.71 ± 1.48 ef	0.78 ± 0.00 g
C3	1.74 ± 0.01 g	26.02 ± 1.33 def	0.96 ± 0.01 ef
C4	1.61 ± 0.01 h	26.16 ± 0.60 e	0.73 ± 0.00 h
BC1	2.51 ± 0.00 a	29.07 ± 1.39 abc	1.34 ± 0.01 d
BC2	2.21 ± 0.01 c	29.05 ± 1.66 abc	0.95 ± 0.00 f
BC3	1.86 ± 0.01 e	29.98 ± 1.43 abc	0.56 ± 0.00 j
BC4	1.98 ± 0.04 d	29.02 ± 1.34 abc	0.71 ± 0.00 i

* Different letters within the same column indicate a significant difference between cultivar × harvest times according to Tukey’s test at the 5% level (*p* < 0.05).

**Table 6 metabolites-12-00798-t006:** Phenolic content found in blueberries (mg/kg frozen fruit).

Samples	Aesculin	*p*-Hydroxybenzoic Acid	5-O-Caffeoylquinic Acid	Caffeic Acid	Aesculetin	Rutin	Quercetin 3-O-glucoside	Ellagic acid	Kaempferol 3-O-glucoside	Quercetin 3-O-rhamnoside	Phloridzin	Quercetin	Kaempferol	Isorhamnetin
D1	0.467 c*	0.000	6.405 d	0.344 c	0.378 b	6.956 a	17.239 b	0.000	0.410 a	0.000	0.309 d	62.995 b	1.032 a	2.857 c
D2	0.661 d	0.000	2.789 ab	0.283 b	0.451 c	6.043 a	17.432 b	0.000	0.471 a	0.000	0.236 c	85.858 cd	1.306 a	5.023 e
D3	1.276 f	0.000	2.191 a	0.566 e	0.630 e	4.780 a	13.985 a	0.000	0.327 a	0.000	0.179 b	119.136 e	1.191 a	6.999 f
D4	0.302 ab	0.000	8.737 f	0.285 b	0.555 d	25.642 d	21.462 c	0.000	5.127 d	0.000	0.091 a	69.045 b	2.409 c	0.752 a
C1	0.455 c	11.048 f	3.746 c	0.709 f	0.833 f	16.367 b	18.261 b	0.708 b	3.392 b	7.318 a	0.143 ab	130.78 f1	2.109 bc	3.927 d
C2	0.277 a	0.970 a	34.793 h	0.318 bc	0.230 a	24.612 cd	22.717 c	0.000	5.767 d	15.452 c	0.159 b	47.120 a	2.042 b	1.127 b
C3	0.297 a	4.693 c	3.241 bc	0.284 b	0.252 a	24.277 cd	24.354 d	0.298 a	7.151 f	12.307 b	0.167 b	90.864 d	2.397 c	0.956 a
C4	0.348 b	6.058 d	7.584 e	0.187 a	0.356 b	22.736 c	22.449 c	0.000	6.575 e	13.912 b	0.137 ab	57.267 ab	1.913 b	0.863 a
BC1	0.463 c	3.407 b	3.202 bc	0.552 e	0.549 d	5.644 a	12.012 a	0.000	0.256 a	0.000	0.338 d	136.391 f	1.109 a	6.905 f
BC2	0.774 e	4.080 c	6.493 d	0.279 b	0.330 b	21.309 c	22.406 c	0.000	5.229 d	12.174 b	0.288 cd	81.966 c	1.996 b	1.471 b
BC3	0.398 bc	8.238 e	9.982 g	0.271 b	0.419 c	26.254 d	23.189 cd	0.000	6.041 e	12.395 b	0.140 ab	72.343 b	2.081 b	0.700 a
BC4	0.440 c	6.702 d	3.589 c	0.449 d	0.569 d	21.425 c	16.110 b	0.000	4.816 c	5.929 a	0.207 bc	92.895 d	2.054 b	1.257 b

* Different letters within the same column indicate a significant difference between cultivar × harvest times according to Tukey’s test at the 5% level (*p* < 0.05).

**Table 7 metabolites-12-00798-t007:** Average content of carbohydrates (g/kg of frozen fruit) in studied blueberry cultivars.

Samples	Sorbitol	Trehalose	Glucose	Fructose	Sucrose	Isomaltose	Turanose	Gentiobiose	Maltose	Maltotriose
D1	3.781 e*	0.019 c	44.471 c	32.659 d	2.267 e	0.077 b	0.088 d	0.002	0.144 d	0.036 ab
D2	1.872 d	0.001 a	39.859 bc	28.124 c	1.564 c	0.003 a	0.125 e	0.002	0.061 b	0.025 a
D3	0.608 ab	0.004 b	24.708 a	15.159 a	1.209 bc	0.002 a	0.038 c	0.001	0.026 a	0.018 a
D4	0.842 b	0.009 a	27.749 a	19.197 a	1.097 b	0.001 a	0.002 a	0.000	0.028 a	0.031 a
C1	0.730 b	0.003 ab	26.864 a	33.680 d	1.844 d	0.002 a	0.002 a	0.000	0.071 b	0.073 c
C2	0.712 b	0.002 a	26.516 a	33.978 d	1.731 d	0.001 a	0.002 a	0.000	0.027 a	0.038 ab
C3	0.485 a	0.002 a	22.424 a	27.080 c	0.922 ab	0.001 a	0.003 a	0.000	0.017 a	0.037 ab
C4	1.044 c	0.004 b	24.586 a	24.908 b	1.015 b	0.001 a	0.003 a	0.001	0.049 ab	0.090 d
BC1	0.412 a	0.001 a	28.632 a	29.396 c	0.682 a	0.002 a	0.005 a	0.000	0.123 c	0.028 a
BC2	0.734 b	0.003 ab	24.809 a	24.993 b	0.850 a	0.003 a	0.008 b	0.000	0.196 e	0.057 b
BC3	1.256 c	0.000	38.287 b	37.820 e	1.023 b	0.002 a	0.003 a	0.000	0.073 b	0.020 a
BC4	0.825 b	0.001 a	36.775 b	31.960 d	1.071 b	0.001 a	0.004 a	0.000	0.126 c	0.329 e

* Different letters within the same columns indicate a significant difference between cultivar × harvest times according to Tukey’s test at the 5% level (*p* < 0.05).

## Data Availability

All data are presented in this manuscript.

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
