# Peer review of "When Is the Right Moment to Pick Blueberries? Variation in Agronomic and Chemical Properties of Blueberry (Vaccinium corymbosum) Cultivars at Different Harvest Times"

_metabolites, 2022, doi:10.3390/metabo12090798_

Round 1
Reviewer 1 Report
metabolites-1833490-peer-review-v1
The paper:
When is the right moment to pick blueberries? Changes in fruit quality of blueberry (Vaccinium corymbosum) cultivars during harvest time
The beginning of the MS idea is very simple.
It is unfortunate that from page No. 7 to Page No. 16 of the MS without numbering the lines.
Unpopular in the abstract abbreviations (RSA), and TPC, TRA in Keywords.
The authors mentioned in line 39 storage capacity, the MS is missing a storage experience that would have been more valuable.
Line 40-41: sentence is incomplete.
Line 46-47 and 49: what is meant by the term storage capacity.
The first and second paragraph in the introduction is too long and can be shortened. It contains unnecessary agricultural information.
Tab. 5 correct to Table 5.
In Table 5 what is meant by % of mature fruits.
Reading the number of fruits for each tree and the yield of trees was useful in this study.
The authors did not mention from which trees the samples were taken, what was the area of the experiment, the number of trees for each cultivar, and what experimental design was used.
4.3. Reagents and standards: It should be reformulated; it contains a lot of unnecessary information.
Why did the researchers use the results of Table 1 LSD ?
Line 136-140:
The conducted ANOVA showed that the significant difference between the cultivars and between harvest dates were determined for almost all studied traits. Is the experiment simple or factorial?. I think that the experiment studies only one factor, due to the different harvest dates. For this picture 1 is incomprehensible.
Line 143-144: remove for three different 143 blueberry cultivars studied.
Line 138: (Fig. 1, 2, 3 and 4) ?
In Table 2 what is the meaning of 1, 2,3 under TPC ?
Line 172: (Figure XA2, XB2 and XC2) ?
2.4. Sugar and sugar alcohol profiles : It should be noted that the data in Table 4.
Table 4: in studier blueberry ? *Different letters in columns indicate a significant difference between cultivar x harvest time according to Tuckey’s test at the 5% level (p < 0.05). ‡The number corresponds to the the score plots in principal component analysis (PCA) (Figure XA2, XB2 and XC2) (?).
In the first paragraph of the discussion the authors used "which was also reported by some other researcers". Effort must be made to interpret the results.
The authors mentioned weather information in the discussion section, may be adding a weather table would be helpful in interpreting the results.
Authors mentioned [44] , 53, 5, 52 ,4,24, 15 1nd 45 stated that, The author's name is mentioned here.
"Sugar alcohols, such as sor-bitol, are very important in human diet since they contribute to gastrointes-tinal effects, including abdominal discomfort, flatus, and diarrhea [56]" Remove it.
[4] stated that the content of total acids in the `Bluecrop` cultivar largely depends on the location where it is grown ?
The conclusions paragraph is too long and can be shortened.
What is the meaning of "four different terms" in the conclusions.
Author Response
Dear Reviewer,
We are very happy with the corrections you have suggested.
It improved our manuscript a lot.
Hopefully You will find our manuscript improved and suitable for publishing in Metabolites.
Document is enclosed
In the behalf of the group of authors,
PhD Milica Fotiric Aksic, associate professor
Department of Pomology, Faculty of Agriculture
University of Belgrade, Belgrade, Serbia

Reviewer 2 Report
The authors have presented a manuscript, which evaluated the agronomic performance of increases and changes in fruit quality of blueberry (Vaccinium corymbosum) cultivars during harvest time. Cultivars were exposed to different right moment to pick blueberries. Visual parameters, weight of blueberries, variation in the content of soluble solids and total acids in fruits and fruit yield were evaluated. The manuscript presents interesting results concerning the selection and the response in the blueberry plant genotypes, but they are some point, need to improve. Following, I have included some comments aimed to enhance the paper:
• The Material and Methods is after Introduction, not after discussion.
• This work presents very interesting results and practice to increase the crop of blueberry cultivars. I think that the authors can improve the format of results demonstration. The authors can highlight better the importance of the results obtained.
• Consider extending the conclusions and adding a Future works paragraph. The summary and Conclusions, it is better to combine them in only section of conclusions.
Finally, the topic of this manuscript is interesting; since the blueberry cultivars during harvest time, but authors must improve the presentation of their material and methods, results and discussion.
Author Response

(The authors gave the same response as above.)

Reviewer 3 Report
Present investigation aimed to determination right moment to pick blueberries and changes in fruit quality 2 of blueberry cultivars during harvest time. After thoroughly evaluation of this study, I found it very interesting and should be considered for the publication. However, before considering it for publication, authors needs to address few minor changes given below
1. Manuscript needs minor language revision
2. Title of the manuscript should be revised
3. Authors are suggested to provide the importance of blueberry in the start of abstract section
4. In the abstract, information should be given about the relationship between important sugars, phenolic compounds, and the harvest time of blueberry fruits.
5. I will suggest the authors to add these studies (https://doi.org/10.1080/15538362.2020.1774476) to support the statement present in lines 64-65.
6. Conclusion section is too long and should be minimized
Author Response
Dear Reviewer,
We are very happy with the corrections you have suggested.
It improved our manuscript a lot.
Hopefully You will find our manuscript improved and suitable for publishing in Metabolites.
Document is enclosed.
In the behalf of the group of authors,
PhD Milica Fotiric Aksic, associate professor
Department of Pomology, Faculty of Agriculture
University of Belgrade, Belgrade, Serbia
